# Social Support and Family Functioning during Adolescence: A Two-Wave Cross-Lagged Study

**DOI:** 10.3390/ijerph19106327

**Published:** 2022-05-23

**Authors:** Xianglian Yu, Xiangtian Kong, Ziyu Cao, Zhijuan Chen, Lin Zhang, Binbin Yu

**Affiliations:** 1School of Psychology, Central China Normal University, Wuhan 430079, China; psyyu@jhun.edu.cn (X.Y.); ziyucao@mails.ccnu.edu.cn (Z.C.); zhijuanchen0513@hotmail.com (Z.C.); 2Department of Education, Jianghan University, Wuhan 430056, China; 3Department of Psychology, University of Essex Colchester, Essex CO4 3SQ, UK; xk21411@essex.ac.uk; 4Key Laboratory of Adolescent Cyberpsychology and Behavior, Ministry of Education, Wuhan 430079, China; 5Key Laboratory of Human Development and Mental Health of Hubei Province, Wuhan 430079, China

**Keywords:** social support, family functioning, adolescent, cross-lagged model

## Abstract

The influence of social and family factors on adolescent mental health has been widely valued. Considering adolescents’ family systems in a broader social context facilitates a better understanding of their mental health, which also has special significance in the post-epidemic era. The purpose of the present study was to explore the relationship between social support and family functioning during adolescence. Students from two middle schools in Fujian province, China, were recruited as participants. Seven hundred and fifty-four participants completed the questionnaire twice in six-month intervals. We constructed a cross-lagged model by using IBM SPSS AMOS 26.0 to test the relationship between these two variables. Social support and family functioning predicted each other in the girls, but not for the boys’ sample. The results of this study suggested that the interaction between family and social factors and the possible gender differences should be considered when dealing with adolescents’ mental health problems.

## 1. Introduction

Adolescent mental health has always been an important issue in psychological practice and research around the world. The World Health Organization (WHO) reported that the prevalence of mental disorders among children and adolescents was 12–29% [1]. In 2021, a cross-national survey in China showed that, overall, 17.6% of children and adolescents had emotional and behavioral problems. Among 12 to 16-year-olds, the prevalence of mental disorders was 19%, with a gradual increase over the past 30 years in China [2]. Adolescent mental health is also related to the well-being of their families and even society, therefore, it is necessary to explore the protective and risk factors of mental health among adolescents.

A previous systematic review demonstrated that both social support and family functioning are psychological protective factors that help individuals adapt and recover from adversity [3]. However, the relationship between social support and family functioning has yet to be scrutinized [3]. The present study focused on social support and family functioning among adolescents by the following justifications: First, adolescence is an age when emotional and behavioral problems are common, such as resistance, loneliness, anxiety, and depression. Second, adolescents socialize more widely, from family to peers, school, and community. This also means that, compared with children, adolescents may seek out more social resources to cope with emotional and behavioral problems [4]. Third, the traditional approach to solving adolescents’ psychological problems is shifting from reducing individual risk factors to enhancing protective factors (such as supportive family and peers) in their lives [5]. Finally, in the post-epidemic era, it is also necessary to study the family and social factors to maintain adolescents’ mental health.

### 1.1. Theoretical Bases

Family systems theory has been widely used in empirical studies on adolescent adaptation in the past decades [6]. According to the family systems theory, the family unit is composed of a series of interrelated relationships formed by different family members, and the family systems are in a dynamic process [7]. Theorists commonly assumed that the family systems should be considered in a broader social context in clinical practice because of the interaction between family and social factors [8]. It was revealed in a previous study that family factors can differentially shape adolescents’ social development [9]. Therefore, according to the family systems theory, we believe that there is an interrelated influence relationship between social support (as a social factor) and family functioning (as a family factor) in adolescents. In addition, the previous study suggested that gender is an important moderator in adolescents’ family relationships and their behavioral problems [6]. Therefore, gender differences also should be considered in the present study.

### 1.2. Social Support

According to The Chinese cultural context, the scholar reclassified the structure of social support, including the three dimensions: subjective support, objective support, and utilization of support [10]. Among the many social support resources, adolescents are most closely connected with peers and family. A meta-analysis showed that supportive peer relationships strongly affected the mental health of adolescents [4]. In addition, family support among different sources of social support had the greatest impact on adolescents’ life satisfaction [11].

Social support as a protective factor for adolescent mental health is highly valued. According to the stress-buffering hypothesis, social support can buffer the negative effects of mental disorders [12] (e.g., anxiety and depression [13], suicidal intentions [14]). Social support is also closely associated with adolescent mental health qualities, such as self-esteem [15] and subjective well-being [16].

### 1.3. Family Functioning

Family functioning refers to the role played by family members in completing various links of family tasks, including the ways of communication and interaction among family members [17]. The presence of a functional family is also associated with positive mental health outcomes [18,19]. More specifically, family cohesion and parental involvement have been tested to have a positive impact on adolescents’ resolution of their current problems [3]. Moreover, adolescents with well-functioning families score higher on emotional intelligence and adolescents’ subjective well-being [20,21].

Numerous studies also demonstrated the negative impact of dysfunctional families on the mental health of adolescents. Poor family functioning not only directly damages teenagers’ mental health but also leads to internalizing and externalizing problems [22] of adolescents through the depression [23] of caregivers [24]. In addition, bad family characteristics (i.e., dysfunctional family, low parental involvement) lead to increased exposure of adolescents to community violence [25,26].

### 1.4. The Relationship between Social Support and Family Functioning

Recent cross-sectional studies also demonstrated that social support can predict family functioning [27,28,29], which indicated that social support from family members is the prerequisite for promoting family functioning [27]. However, a few cross-sectional studies have found that family functioning positively predicts social support [30,31]. To our knowledge, the relationship between social support and family functioning is unclear. Therefore, it is important to test the relationship between them by examining the time series [19]. What is more, some researchers found that traditional Chinese culture emphasizes family over the individual [32], therefore, family functioning in Chinese culture may also be different from those in western families.

### 1.5. The Current Study

Overall, with middle school students in southeast China as the participants of this study, the interaction relationship between social support and family functioning during adolescence was explored through two waves of longitudinal studies. The purpose of this study was to provide evidence for the interrelated influence relationship between adolescents’ family functioning and social factors in family systems theory and to provide practitioners and researchers with a better understanding of the interaction between social and family factors in adolescent psychological development. According to the theoretical basis of the present study, we believe that social support and family functioning are influenced by each other, while gender may be an important moderator.

## 2. Method

### 2.1. Participants

Participants were recruited from two high schools in southeast China by convenient sampling. The current study included scores for social support and family functioning among participants at both schools. Students included in this study were in grades 7 through 11 (ages 12–19; M = 15.03, SD = 1.56). A total of 754 samples were collected, of which 320 (42.4%) were boys and 434 (57.6%) were girls.

### 2.2. Procedures

Before the study officially began, we contacted the principals of both schools in advance to explain the purpose of the study. After seeking the schools’ consent, we issued informed consent to students. Participants were aware that they were free to withdraw from the study at any time. All the data were collected in line with a strictly voluntary and anonymous basis. All data would be used only for research. All study procedures were approved by the Life Science Ethics Committee of Central China Normal University.

In April 2021, we conducted the first wave of a questionnaire survey in the form of an online questionnaire (time 1). Then, in September 2021, we collected the data from the second wave of the online survey (time 2), with an interval of 6 months. A total of 1230 students participated in the survey. After removing invalid questionnaires, 754 valid questionnaires were finally included. The criteria for eliminating data were as follows: (1) The filling time was more than 20 min or less than 1 min; (2) the total score of these two variables was greater than 3 standard deviations; (3) the missing values in the questionnaires were more than 10%.

### 2.3. Measures

#### 2.3.1. Social Support

The Social Support Rating Scale questionnaire (SSRS) [10] consists of 10 items and three dimensions: objective support (e.g., “In the past, when you encounter a difficult situation, what have been the sources of comfort and care?”) subjective support (e.g., “How many friends do you have to help and support you?”), and support utilization (e.g., “What is your way of asking for help when you are in trouble?”) In the present study, the total score of social support was used to represent the level of social support of participants. Internal consistencies of the mean scale scores were acceptable at time 1 (and time 2): α = 0.67 (0.64).

#### 2.3.2. Family Functioning

The Chinese version of the General Functioning Scale (GF-FAD-C), based on the General-Functioning Subscale of the Family Assessment Device [33] was used to measure family functioning. The GF-FAD-C has 12 items to assess the overall health/pathology of the family (e.g., “It is difficult to arrange some family activities because of the misunderstanding between us”). To facilitate the understanding, reversed GF-FAD-C scores were used to represent family functioning. In the present study, internal consistencies of the mean scale scores were acceptable at time 1 (and time 2): α = 0.80 (0.85).

### 2.4. Statistical Analyses

Statistical analyses in the present study were conducted using IBM SPSS Statistics for Windows, Version 26.0, IBM SPSS AMOS 26.0. First, descriptive and correlation analyses were conducted for the two variables. According to our hypothesis, the cross-lagged panel model (CLPM) was then used to construct a structural equation model to investigate the interrelated influence relationship between social support and family functioning in the sample of adolescents. Then, we also constructed cross-lagged models in the samples of boys and girls, respectively. In this way, it is possible to compare whether there were gender differences in the relationship between adolescents’ social support and family functioning.

## 3. Results

### 3.1. Descriptive Analyses and Correlation Analyses

Participant attributes are shown in Table 1. Intercorrelations among study variables and descriptive statistics are provided in Table 2. The descriptive statistics of social support, family functioning at time 1 (M = 38.7, SD = 7.3; M = 36.44, SD = 5.30), and social support, family functioning at time 2 (M = 39.58, SD = 7.32; M = 36.37, SD = 5.01) can be seen in Table 1. There was a significant positive correlation between the two measures of social support (r = 0.40) and family functioning (r = 0.47). These two variables were significantly intercorrelated at time 1 and time 2 (r = 0.39, 0.37; *p* < 0.01).

### 3.2. Cross-Lagged Model of Social Support and Family Functioning

In the present study, we constructed a manifest variable cross-lagged model to examine the relationship between social support and family functioning (see Figure 1). As expected, stability paths were all significant from time 1 to time 2. The autoregressive path coefficient for social support and family functioning from time 1 to time 2 were quite small (social support β = 0.37, SE = 0.036, *p* < 0.001; family functioning β = 0.41, SE = 0.033, *p* = 0.001), indicating that there were notable changes in these two variables between the two measurement occasions.

In the next step, the cross-lagged model examined whether changes in family functioning were related to social support. As can be seen in Figure 1, the cross-path coefficient from social support at time 1 to family functioning at time 2 was statistically significant (β = 0.14, SE = 0.024, *p* < 0.001). Considering the other direction of the relationship, the cross-path from family functioning at time 1 to social support at time 2 was also statistically significant (β = 0.08, SE = 0.05, *p* < 0.05). Taken together, these results lend some support for bidirectional relations between the constructs across a period of 6 months.

We then examined the relationship between social support and family functioning in the sample of different genders. The results for girls can be seen in Figure 2, where the cross-path coefficient from social support at time 1 to family functioning at time 2 was statistically significant (β = 0.20, SE = 0.033, *p* < 0.001). The cross-path from family functioning at time 1 to social support at time 2 was also statistically significant (β = 0.16, SE = 0.066, *p* < 0.001). However, the cross-lag model results of the boys’ sample were completely different from those of the girls’ sample, as can be seen in Figure 3. The boys’ cross-path coefficients for social support and family functioning were not statistically significant in the 6 months interval.

## 4. Discussion

The present study constructed a cross-lagged model to investigate the relationship between social support and family functioning among Chinese adolescents. Some valuable results have been found in the present study. First, the result showed that social support could predict family functioning positively, which is consistent with previous cross-sectional studies [27,29], When individuals face traumatic events, social support can play a protective role and has positive significance for mental health [13]. From this perspective, we should attach importance to social support for coping with dysfunctional families.

In terms of predicting social support from family functioning, the longitudinal study had shown that personality and family climate factors predicted social support, with family cohesiveness playing a significant role [34]. It is true, therefore, that adolescents experience more emotional well-being in a normal family system where family members can rely on social support [30].

The bidirectional predictive relationship between social support and family functioning was found in the present study, which can be supported by a previous study [35]. The possible reason is that support from family members is still an important social resource, although the focus of adolescence gradually transfers from the family to the outside world. In addition, cross-cultural studies found that Chinese people are more likely to get help from family before external social support [36].

Furthermore, it is interesting to find significant gender differences in the longitudinal relationship between social support and family functioning. Social support and family functioning predicted each other in the girls, but not for the boys’ sample. Gender differences are common in studies of adolescent family systems [37,38,39]. For example, harmonious family communication reduces emotional and behavioral problems only among adolescent girls [40]. The possible explanation is that, compared with boys, girls are highly sensitive to the family, and they are more easily enmeshed in the family [41]. Adequate social support also enables girls to acquire better interpersonal skills, which can facilitate their family communication [30]. From another perspective, in dysfunctional families, girls are more likely to have social anxiety, which may limit their utilization of social support [42]. For boys, when there is a lack of good family communication, they are more likely to have externalizing behavior problems rather than actively seek social support [43]. This also indicates that boys avoid family conflict by generating externalizing behavioral problems, which may also explain the absence of an interactive relationship between social support and family functioning in the boys’ sample [6].

There are several strengths in the present study. First, the present research explored the relationship between social support and family functioning based on family systems theory. The results of this study provide evidence for the interaction between family systems and adolescents’ social factors in this theory. Second, to the best of our knowledge, a limited number of longitudinal studies have examined the relationship between the two variables. Therefore, the present study aims to test the relationship between social support and family functioning through a cross-lagged model. Lastly, the relatively large sample enhances the credibility of the study.

Our findings may have several implications. Theoretically, the present study provides evidence for family systems theory. We attempt to place the family systems of adolescents’ development in a larger social context and explore the influence between adolescents’ social support and family functioning. Moreover, the results of this study also provide a basis for personnel involved in adolescent mental health intervention with adolescents. Currently, some mindfulness-based interventions provided us with reference to improve the mental health of parents and children and improve family functioning [44]. Another meta-analysis showed that interpersonal psychotherapies (e.g., adolescent skills training and attachment-based family therapy) in school were increasingly proving to be effective treatments for depression [30]. Future intervention studies can be carried out to improve the family functioning and social support level of adolescents, and gender differences should be paid attention to in the intervention [45].

It is meaningful to investigate the protective factors of adolescent mental health in the post-epidemic era. Adolescents’ emotional support and social development heavily rely on social connections [46]. However, due to the impact of the epidemic, adolescents were isolated from their friends and other social support networks, which can bring potential psychological risks [47]. At the same time, social restrictions have led teens to spend more time with family members. Previous studies have shown that adolescents who were accompanied by a parent also reported significantly lower levels of depression and anxiety than adolescents who stayed home alone all day [41], which suggested that support from the family can help to compensate for the lack of social connection available in the lockdown period [47]. Therefore, in the context of COVID-19, we further believe in the positive role of adolescents’ social support systems and family function in adolescents’ psychological adjustment.

There were still some limitations that need to be improved in future studies. First, self-report questionnaires were used in our study; therefore, the results of the questionnaire may be affected by participants’ subjective experiences. Second, in the present study, the relationship between these two variables in the latent variable model was not tested. Third, the cross-lagged model of the current study only collected two data sets. A third data set could be collected in the future to further enhance the credibility of the longitudinal study. The results of this study can also inspire us to conduct follow-up research. As far as we know, social support and family functioning are closely related to adolescent traumatic experiences, depression, and other psychological problems. Therefore, other variables related to emotional problems can be included in future longitudinal studies to test their relationship. Furthermore, the influence of individual factors can be further investigated (e.g., personality, cognitive pattern).

## 5. Conclusions

The present study shed new light on the relationship between social support and family functioning by constructing a cross-lagged model. Based on our preconceived hypothesis, our results demonstrate the importance of the interaction relationship between social support and family functioning among adolescents, as well as the existence of gender differences. It is necessary to design intervention studies for adolescents of different genders. The effects of future intervention studies to enhance adolescents’ social support levels could be simultaneously transferred to improve their family functioning, and vice versa.

## Figures and Tables

**Figure 1 ijerph-19-06327-f001:**
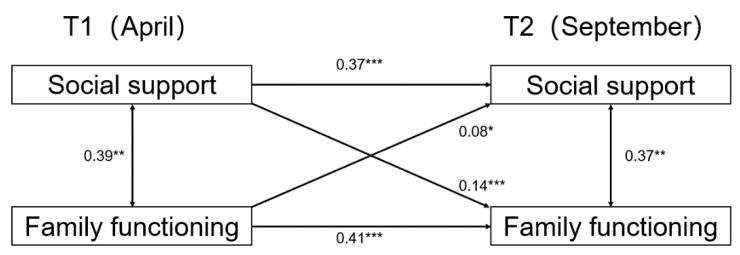
Cross-lagged model for social support and family functioning for the total sample. Note: statistically significant standardized path coefficients are shown with solid lines. * *p* < 0.05. ** *p* < 0.01. *** *p* < 0.001.

**Figure 2 ijerph-19-06327-f002:**
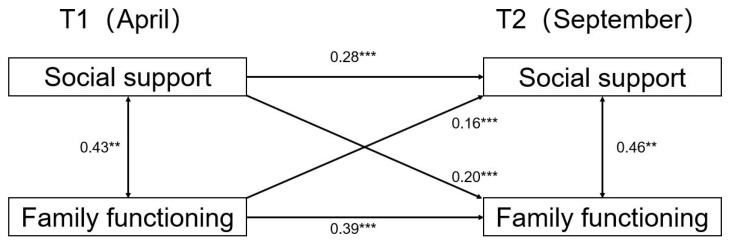
Cross-lagged model for social support and family functioning for girls. Note: statistically significant standardized path coefficients are shown with solid lines. ** *p* < 0.01. *** *p* < 0.001.

**Figure 3 ijerph-19-06327-f003:**
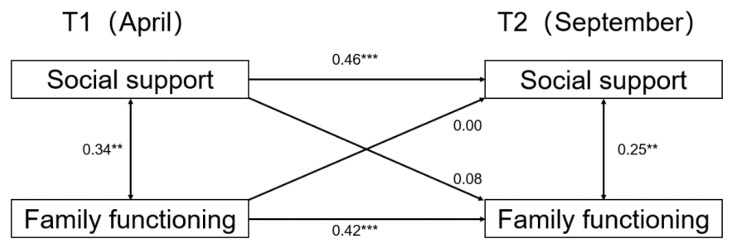
Cross-lagged model for social support and family functioning for boys. Note: statistically significant standardized path coefficients are shown with solid lines. ** *p* < 0.01. *** *p* < 0.001.

**Table 1 ijerph-19-06327-t001:** Participant attributes in the present study.

Participant Attributes	N (%)	Mean (SD)
Gender		
Male	320 (42.4%)	
Female	434 (57.6%)	
Age		15.03 (1.56)
Grade		
7th	286 (37.9%)	
8th	176 (23.3%)	
10th	127 (16.8%)	
11th	165 (21.9%)	

**Table 2 ijerph-19-06327-t002:** Descriptive and correlation analysis.

	1	2	3	4	5	6
1. Sex	——					
2. Age	0.068	——				
3. T1 Social support	−0.090 *	−0.167 **	——			
4. T1 Family functioning	−0.043	−0.086 *	0.389 **	——		
5. T2 Social support	−0.091 *	−0.069	0.405 **	0.228 **	——	
6. T2 Family functioning	−0.039	−0.169 **	0.301 **	0.465 **	0.367 **	——
7. M		15.03	38.7	36.44	39.58	36.37
8. SD		1.56	7.37	5.3	7.32	5.01

Note. For sex, 1 = boys, 2 = girls. T1 = time 1; T2 = time 2 (approximately 6 months after time 1). * *p* < 0.05. ** *p* < 0.01.

## Data Availability

The data sets analyzed for the current study are available in the American Educational Research Association Data Repository of the open ICPSR Repositories at the University of Michigan Interuniversity Consortium for Political and Social Research (ICPSR): http://doi.org/10.3886/E164182V2.

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
