# Peer review of "Social Support and Family Functioning during Adolescence: A Two-Wave Cross-Lagged Study"

_ijerph, 2022, doi:10.3390/ijerph19106327_

Round 1
Reviewer 1 Report
In the conclusion it mentions hypothesis but in the methodology it is not mentioned that there is any hypothesis
Author Response
We are grateful for taking your time to review this manuscript,our response is as follows.

Reviewer 2 Report
I have no further comments
Author Response
We are grateful for taking your time to review this manuscript. Thank you very much for your positive feedback on our manuscript. We will also carry out relevant follow-up studies.
Reviewer 3 Report
I am going to assess the following revision of the article, based on the personal observations noted above.
First of all, he made some observation in relation to the abstract. It indicated that the background could be further expanded and the methods added more explicitly and not only with the treatment applied. In this regard, the authors of the manuscript have added two sections: In the first one, they highlighted "the influence of social and family factors on the mental health of adolescents", pointing out that "it has already been widely valued". They then stated that, in a broader, post-pandemic social environment, adolescents' family systems facilitate a better understanding of their mental health. Secondly, they expressed the results of the study regarding "the interaction between family and social factors and possible gender differences", considerations that must be taken into account when we talk about adolescent mental health problems.
I consider that the background is further developed by adding the corresponding citations and is completed by highlighting the interrelationship that occurs between family and social factors and possible gender differences.
In the introduction, it indicated that some element was missing that clarifies a little more the meaning of the objective. In this sense, the authors add: "to provide evidence of the interrelated influence relationship between adolescent family functioning and social factors in family systems theory." About completing with some main conclusion, they indicate that "social support and family functioning mutually influence each other, while gender can be an important moderator". This covers the requirement.
In the discussion, I meant that the findings and their implications needed to be further discussed from a broader context. On this, four new elements are provided with their corresponding current references.
The relationship with the proposed hypothesis has been added in the conclusion, demonstrating the importance of the interaction between social support and family functioning among adolescents and the existence of different genders, aspects that have been reiterated as similar additions in several sections. Also, the determination of future research from the current prospective is specified in the approach of future intervention studies to improve the levels of social support of adolescents to improve their family functioning, and vice versa.
As indicated, and as far as I am concerned, the authors have provided the necessary elements and the evidence that complete the perceived deficiencies.
Author Response
We are grateful for taking your time to review this manuscript. As you have observed, the manuscript we have resubmitted is of much higher quality than the previous one. Thank you very much for your positive feedback on our revised manuscript. We will also carry out relevant follow-up studies.
This manuscript is a resubmission of an earlier submission. The following is a list of the peer review reports and author responses from that submission.
Round 1
Reviewer 1 Report
Try to be more described with the results, the tables can be separated and with more description before each one because lost the information.

Author Response
Thank you for your valuable comments. Our point-by-point response is shown in the file.

Reviewer 2 Report
The authors present their results on Social Support and Family Functioning during Adolescence: A 2 Two-Wave Cross-Lagged Study . It is an interesting manuscript, which could be improved after revision.
Abstract section
Abstract: provides a fair summary of the article
-"The purpose of the present study was to explore the causal relationship between social support and family functioning during adolescence"
The word causal should be deleted the keep the word relationship or interaction
Introduction Section
Introduction: The introduction is clear, useful in setting the scene and understandable but there are few, minor, errors in sentence construction. Moreover introduction section is very extended comparing to the length of the study
Materials and methods section :
-"Participants were recruited from two high schools in southeast China by convenient sampling" What is the meaning of " by convenient sampling"
-Why a filling time more than 20 minutes was one of the exclusion criteria?
-Why a score difference between the two tests greater than 2 standard deviations was one of the exclusion criteria? In such a case I think that the relationship or the interaction of the two parameters is supported by the design of the study. Otherwise please explain better your exclusion criteria.
Results section
-An analysis by sex and/or age would be also interesting to be performed
Author Response

(The authors gave the same response as above.)

Reviewer 3 Report
The article studies the causal relationship between social support and family run during adolescence. It highlights the importance and influence between both, regarding the mental health of adolescents. The topic is current and interesting for the comprehensive development of adolescents, especially in this post-pandemic era. Regarding the manuscript, I present the following observations, which I will later summarize in the evaluation report.
In the abstract, the background could be further expanded, and also adding explicit methods, not only the treatment applied.
The introduction highlights the importance of the topic, the purpose of the research, the current state of the object of study, important publications and the hypothesis are cited, but I consider that something is missing that could clarify a little more the meaning of the objective and add some main conclusion.
The method adequately and sufficiently describes the elements that allow a subsequent replication. Likewise, the participants, the procedures, the instruments used and the exclusion criteria are indicated, as well as the associated protocols followed and the instrument used to collect information.
The results are shown through the profile of the students in Table 1, with the descriptive data in Table 2. The significant positive correlation between the variables of social support and family functioning is also indicated. The cross-variable model explores the relationship between social support and family functioning. The results clarify the interrelation between both variables during the applied period.
The discussion offers the results interpreted from the perspective of previous studies with adolescents. The limitations of the study are adequately indicated. However, I believe that the findings and their implications should be further discussed from the widest possible context and should further indicate the determination of future research from the current perspective. It would also be necessary to discuss more from the proposed hypothesis.
The references, although they present a self-citation, do not represent any difficulty, being adequate and sufficient.
Author Response

(The authors gave the same response as above.)
